# Nitrogen Limitation of Intense and Toxic Cyanobacteria Blooms in Lakes within Two of the Most Visited Parks in the USA: The Lake in Central Park and Prospect Park Lake

**DOI:** 10.3390/toxins14100684

**Published:** 2022-10-06

**Authors:** Jacob M. Flanzenbaum, Jennifer G. Jankowiak, Jennifer A. Goleski, Rebecca M. Gorney, Christopher J. Gobler

**Affiliations:** 1School of Marine and Atmospheric Sciences, Stony Brook University, New York, NY 11794, USA; 2Division of Water, New York State Department of Environmental Conservation, Albany, NY 12233-0001, USA

**Keywords:** microcystin, harmful algal bloom, cyanobacteria, nutrient limitation

## Abstract

The Lake in Central Park (LCP) and Prospect Park Lake (PPL) in New York City (NYC), USA, are lakes within two of the most visited parks in the USA. Five years of nearshore sampling of these systems revealed extremely elevated levels of cyanobacteria and the toxin, microcystin, with microcystin levels averaging 920 µg L^−1^ and chlorophyll *a* from cyanobacterial (cyano-chl*a*) populations averaging 1.0 × 10^5^ µg cyano-chl*a* L^−1^. Both lakes displayed elevated levels of orthophosphate (DIP) relative to dissolved inorganic nitrogen (DIN) during summer months when DIN:DIP ratios were < 1. Nutrient addition and dilution experiments revealed that N consistently limited cyanobacterial populations but that green algae were rarely nutrient limited. Experimental additions of public drinking water that is rich in P and, to a lesser extent N, to lake water significantly enhanced cyanobacterial growth rates in experiments during which N additions also yielded growth enhancement. Collectively, this study demonstrates that the extreme microcystin levels during blooms in these highly trafficked lakes represent a potential human and animal health threat and that supplementation of these artificial lakes with public drinking water to maintain water levels during summer may promote the intensity and N limitation of blooms.

## 1. Introduction

Cyanobacteria are a phylum of prokaryotic bacteria that use photosynthesis to obtain energy and are an important component of freshwater microbial communities. In freshwater ecosystems, several genera of cyanobacteria can form dense ‘blooms’ that can indirectly cause deleterious effects on the surrounding environment, such as hypoxia, and can directly pose threats due to the production of a wide suite of toxins [1]. The most wide-spread and well-studied of the cyanotoxins are hepatotoxic cyclic peptides, with the most common being the microcystins [2]. Other common cyanotoxins include neurotoxic alkaloids such as anatoxins and saxitoxins and the polycyclic uracil derivative, cylindrospermopsin which is a gastrointestinal toxin [1]. Not all cyanobacteria produce toxins, and even within a toxin-producing genera, not all individuals will contain the genes necessary for toxin biosynthesis [3]. Cyanotoxins, however, are of concern, as cyanobacterial toxins such as microcystins and anatoxins have been linked to both human and animal poisoning events [1,4], with human health impairment associated with skin allergies, inhalation of aerosol containing cyanobacterial cells, and consumption of drinking water tainted with cyanotoxins [4].

Many factors influence the dominance of cyanobacterial harmful algal blooms (CHABs) with anthropogenic nutrient loading as a prime promoter [5]. A clear link exists between intensification of CHABs and anthropogenic nutrient loading [6,7]. Phosphorus (P) has traditionally been considered the main driver of cyanobacterial blooms, with P loading demonstrated to increase levels of primary production and the relative abundance of cyanobacteria in freshwater systems [8,9]. In recent years, however, the dominance of non-diazotrophic phytoplankton species including *Microcystis* and *Planktothrix* has often been attributed, in part, to nitrogen (N) loading in freshwater systems [10,11,12]. The relationship between blooms of cyanobacteria and elevated levels of nutrients has long been established, but only recently has a positive relationship between eutrophic conditions and an increase in dominance of toxin-producing cyanobacteria strains over their non-toxic counterparts been demonstrated [13,14,15].

In the last decade, putative CHABs have been observed in The Lake in Central Park (LCP) in Manhattan, NY, USA, and Prospect Park Lake (PPL) in Brooklyn, NY, USA (Appendix A), during the summer months. The parks around these lakes are well-trafficked locations, receiving more than 40 million visitors annually, and are surrounded by the densest metropolitan area in the US. Both LCP and PPL are popular recreational, fishing, and boating destinations with lake-side trails surrounding the lakes. Despite the potential risk CHABs represent within these parks, no study to date has described nor quantified the phytoplankton community composition and succession, measured levels of cyanobacteria, cyanotoxins, or nutrients, or determined what nutrients limit phytoplankton community growth in either lake.

The goal of this study was to characterize CHABs in LCP and PPL and to evaluate the role of N and P in promoting these events. From 2015 to 2019, the levels and types of cyanobacteria were assessed along with microcystin concentrations. In 2020, the seasonal dynamics of the phytoplankton community assemblages, pools of nitrogenous and phosphorus containing nutrients, concentrations of microcystin and concentrations of microcystin and saxitoxin synthesis genes (*mcyE* and *sxtA*, respectively) in the LCP and PPL were quantified from June through October. In addition, experiments were performed to assess whether N, P, a combination of both elements, or neither were limiting to the various phytoplankton populations present in each lake. Understanding the species composition and toxicity of the blooms in relation to nutrients that may be limiting in these systems could aid in the development of management strategies to limit the public health risk associated with CHABs in LCP and PPL.

## 2. Results

Shoreline cyanobacterial dynamics, 2015–2019—The concentrations of cyanobacteria in both LCP and PPL were consistently above bloom levels, the threshold of which is set at 25 μg L^−1^ by the DEC [16], during the monitoring period of 2015–2019 (Figure 1, Figure 2). In LCP, the mean seasonal cyanobacterial biomass was 1.21 ± 0.62 × 10^4^ μg L^−1^ in 2015, 3.16 ± 0.92 × 10^4^ μg L^−1^ in 2016, 2.04 ± 0.86 × 10^4^ μg L^−1^ in 2017, 1.68 ± 0.89 × 10^4^ μg L^−1^ in 2018, and 1.12 ± 0.76 × 10^5^ μg L^−1^ in 2019 (Figure 1). PPL had seasonal averages of 2.18 ± 0.78 × 10^4^ μg L^−1^ in 2015, 3.43 ± 1.46 × 10^4^ μg L^−1^ in 2016, 3.56 ± 1.52 × 10^5^ μg L^−1^ in 2017, 2.09 ± 1.1 × 10^5^ μg L^−1^ in 2018, and 1.31 ± 0.53 × 10^5^ μg L^−1^ in 2019 (Figure 2). Throughout the study of LCP and PPL, cyanobacteria dominated phytoplankton biomass making up more than 99% of total phytoplankton biomass from 2015 to 2019 (Table 1). *Microcystis* was the most commonly dominant cyanobacterium, with *Planktothrix, Dolichospermum,* and *Aphanizomenon*. In LCP *Microcystis* was identified in 100% of samples taken and was the sole identifiable cyanobacterium in in 64% of samples (Appendix A), whereas, in PPL *Microcystis* was identified in 91% of samples and was the sole identifiable cyanobacterium in only 14% (Appendix A).

Shoreline microcystin concentrations, 2015–2019—Levels of microcystin in the lake varied throughout the monitoring period but were detected in nearly every sample analyzed from 2015 to 2019 (100% in LCP, 98% in PPL). In LCP toxin concentrations averaged 2.73 ± 1.58 × 10^3^ μg L^−1^ in 2015, 2.32 ± 0.75 × 10^3^ μg L^−1^ in 2016, 7.34 ± 2.09 × 10^2^ μg L^−1^ in 2017, 1.09 ± 0.27 × 10^3^ μg L^−1^ in 2018, and 7.70 ± 3.42 × 10^2^ μg L^−1^ in 2019 (Figure 1). PPL microcystin concentrations averaged 1.51 ± 0.48 × 10^3^ μg L^−1^ in 2015, 1.52 ± 0.45 × 10^3^ μg L^−1^ in 2016, 7.75 ± 1.22 × 10^2^ μg L^−1^ in 2017, 5.51 ± 1.76 × 10^2^ μg L^−1^ in 2018, and 43 ± 12 μg L^−1^ in 2019 (Figure 2).

### 2.1. Field Monitoring, 2020

The Lake in Central Park—Over most of the summer (June 17–Sept 9) temperature levels remained fairly consistent (25.8 ± 1.7 °C), before dropping to ~17 °C by September 21 and remaining around that level for the rest of the season (Figure 3a). Nutrient levels in LCP were low in June 2020 with NO_x_ at 1.25 ± 0.22 μM, NH_4_^+^ at 0.96 ± 0.32 μM, and PO_4_^−3^ at 0.53 ± 0.15 μM (Figure 3b,c). NH_4_^+^ concentration remained low throughout the season, averaging 0.86 ± 0.39 μM, and NO_x_ averaged 2.07 ± 2.3 μM, with a peak on 7/15/20 (7.98 ± 1.5 μM; Figure 3b). PO_4_^−3^ levels slowly rose to a peak of 5.81 ± 3.63 μM in the middle of the season (8/12/2020) and remained around these levels until slowly declining by the end of sampling (10/22/2020) (Figure 3c). Total nitrogen (TN) and total phosphorus (TP) levels generally paralleled cyanobacterial biomass and averaged 172.3 ± 65.9 µM and 10.2 ± 3.5 for the year, respectively (Figure 3b, c). Ratios of dissolved inorganic nitrogen to dissolved inorganic phosphorus ranged from 1.3–5 μM during June and July but declined to <1 for August, September, and October (Figure 3d). TN:TP ratios were relatively consistent throughout the season, ranging from 13–18 (Figure 3d).

During initial sampling, green algae were dominant over cyanobacteria, making up around 95% of biomass sampled (Figure 4a). However, cyanobacteria rapidly increased as lake temperatures rose, and by 7/15/2020 comprised 97% of biomass and remained dominant until October, when concentrations dropped to levels similar to green algae (Figure 4a). Cyanobacteria reached peak biomass in mid-summer (214 μg L^−1^ on 8/26/2020) and averaged 92 ± 67 μg L^−1^ throughout the season (Figure 4a). *Microcystis* was the dominant genera in every sample analyzed, and was the sole genera identified in 60% of samples, with *Dolichospermum*, *Planktothrix*, and *Aphanizomenon* identified in substantially lower quantity (Appendix A).

The saxitoxin biosynthesis gene, *sxtA,* concentration was low for most of the year and below detection during initial sampling but rose to a peak concentration of 2.26 ± 0.26 × 10^6^ copies mL^−1^ on 7/15/20 co-occurring with the rise in cyanobacterial biomass in the lake (Figure 4b). Thereafter, *sxtA* concentrations average of 2.3 ± 3.2 × 10^4^ copies mL^−1^ for the remainder of the season, with the gene undetectable on some dates (Figure 4b). In contrast, the microcystin producing gene *mcyE* was detectable throughout the sampling season. Initially at a low concentration (1.15 ± 0.26 × 10^4^ copies mL^−1^), *mcyE* concentration rose to have three distinct peaks throughout the sampling period, 7/15/2020 (5.13 ± 1.76 × 10^5^ copies mL^−1^), 8/12/20 (4.05 ± 0.42 × 10^5^ copies mL^−1^), and 9/21/2020 (8.22 ± 5.56 × 10^5^ copies mL^−1^) (Figure 4b). Overall, the average concentration of *mcyE* for the season was 2.63 ± 2.62 × 10^5^ copies mL^−1^. Microcystin concentrations were low during the first month of sampling, on average 0.83 ± 0.43 μg L^−1^ and in parallel with *mcyE* concentrations, microcystin levels had a small initial peak of 14.2 μg L^−1^ on 7/15/2020 early in the season, rose to a second peak of 24.9 μg L^−1^ in August and remained at an average of 23.4 ± 1.37 μg L^−1^ from 8/26/2020 to 9/21/2020 before decreasing in October (Figure 4c).

Prospect Park Lake—Temperature remained high during the initial months of sampling, fluctuating around an average of 28 °C from 6/24/2020 to 7/22/2020, and then declining through the rest of the season to a low of 15 °C on 10/29/2020 (Figure 5a). Unlike LCP, in PPL lake nutrient levels were initially elevated in June and then decreased (Figure 5b). NH_4_^+^ was 26.2 ± 0.57 μM June then dropped below 5 μM and averaged 1.35 ± 0.77 μM from 8/5/2020 to 10/30/2200 (Figure 5b). NO_x_ remained low throughout the entirety of the 2020 season, averaging at 1.92 ± 0.9 μM (Figure 5b). TN levels were also measured at their highest levels upon initial sampling (283.4 ± 46.3 μM) before trending downward for the rest of the season (Figure 5b), with PO_4_^−3^ and TP followed nearly identical trends, starting at 8.09 ± 0.22 μM and 15.5 ± 0.37 μM, respectively, (Figure 5c). DIN:DIP began at 2.4:1, dropped to a low of 0.4:1 mid-season (8/19/2020), ended at the highest ratio 4:1 DIN:DIP (10/30/20) and averaged 1.0 ± 1.2 for the sampling season (Figure 5d). TN:TP ranged from 0.4 to 17 and averaged 11.5 ± 3.4 for the sampling season (Figure 5d).

In early June, concentrations of cyanobacteria and green algae were similar, 7.7 ± 0.6 μg L^−1^ and 10.9 ± 0.18 μg L^−1^, respectively (Figure 6a). By the second week of sampling (6/25/2020), green algae concentrations decreased below detectable levels, and cyanobacterial biomass increased to 139 ± 0.8 μg L^−1^ and remained dominant for the rest of the sampling season (Figure 6a). Cyanobacteria had a second peak in September, averaging 156 ± 5.1 μg L^−1^ that month, before dropping again and averaging 53.8 ± 5.9 μg L^−1^ for October (Figure 6a). Of the cyanobacterial genera identified, *Microcystis* was present on every date and was the sole identifiable genera in 40% of samples (Appendix A). *Planktothrix* appeared in 50% of samples, and *Aphaninizomenon*, and *Dolichospermum* were detected intermittently (Appendix A). 

Both *sxtA* and *mcyE* concentrations were one-to-two orders of magnitude lower in PPL compared to LCP. Both genes were below the detection limit in PPL in June. Levels of *sxtA* rose to 4.2 ±0.4 × 10^5^ copies mL^−1^ in July and to 3.03 ± 0.25 × 10^5^ copies mL^−1^ in September before decreasing to 2.06 ± 1.06 × 10^3^ copies mL^−1^ on 9/16/20 then decreasing to below detection limits on 10/28/2020 (Figure 6b). Levels of *mcyE* steadily increased to 4.15 ± 0.77 × 10^3^ copies mL^−1^ on 8/5/2020, briefly decreased, and then rose to 2.36 ± 0.51 × 10^4^ copies mL^−1^ on 9/2/2020. Thereafter, *mcyE* levels remained low until the end of the season (Figure 6b). Microcystin concentration were 0.28 μg L^−1^ in June, were below detectible levels during summer and then increased in September when concentrations were 0.47 μg L^−1^, with concentrations below detectible levels thereafter (Figure 6c).

### 2.2. Nutrient Amendment Experiments Using Drinking Water and Nutrients, 2018

The Lake in Central Park—During the first experiment, the addition of 5 µM N (as nitrate) significantly increased cyanobacterial levels over the control (Figure 7a, *p* < 0.05), while the addition of drinking water caused a larger increase (Figure 7a, *p* < 0.001). In the second experiment (Figure 7b), four treatment groups were found to have significantly larger cyanobacterial biomass relative to the control: the addition of nitrate (15µM) (*p* < 0.05), the addition of drinking water (*p* < 0.01), and the addition of N + P at high and low concentrations (*p* < 0.01). In the final experiment, the additions of drinking water (*p* < 0.001) and nitrate (15 µM; *p* < 0.05) yielded significantly higher cyanobacterial biomass relative to the control (Figure 7c).

Prospect Park Lake—In two experiments, nitrate treatments consistently resulted in an increase in cyanobacterial biomass relative to the control regardless of concentration. In both experiments, the two nitrate treatments (5 µM and 15 µM) caused the largest positive, significant increase in cyanobacterial biomass relative to the control treatments (Figure 8a,b; *p* < 0.0001 for all). In addition, during the first experiment, the drinking water treatment and P additions caused significant but smaller increases in cyanobacterial biomass Figure 8a, (*p* < 0.01).

### 2.3. Nutrient Amendment Experiments, 2020

The Lake in Central Park—The nutrient amendment experiments were conducted to determine whether nitrogen, phosphorus, or both nutrients would impact algae community composition and growth. The nitrogen and nitrogen + phosphorus amendment treatments yielded a significant increase in cyanobacterial biomass relative to the control in 80% of the whole water experiments and in 100% of the nutrient dilution experiments (Figure 9). Green algae, on the other hand, displayed significantly higher biomass in the nitrogen and nitrogen + phosphorus treatments in 30% of the whole water experiments and 20% of dilution experiments conducted (Figure 9). Phosphorus alone had no impact on growth of cyanobacteria or green algae in any experiment conducted in LCP.

Prospect Park Lake—In the PPL experiments, nitrogen treatments lead to a significantly higher biomass nine out of ten of whole water experiments and half of the nutrient dilution experiments (*p* < 0.05 for all; Figure 10). Additionally, in one of ten whole water experiments (7/8/2020), the phosphorus addition yielded a significant increase in biomass, along with nitrogen and nitrogen + phosphorus (Figure 10b, *p* < 0.001). The nitrogen treatment groups consistently had an identical impact on growth when compared to the nitrogen and phosphorus treatments. No nutrient treatment in any of the 10 whole water experiments or 10 dilution experiments lead to a significant change in the levels of green algae (Figure 10).

## 3. Discussion

During this study, extremely elevated levels of cyanobacteria and the toxin, microcystin, were found annually in the near shore environment of lakes within two of the most visited public parks in the United States, Central Park and Prospect Park of New York City. Levels of microcystin in nearshore regions ranged from ~1 to 7,500 µg L^−1^ and averaged 920 µg L^−1^, representing a significant potential human and animal health risk given the occurrence of boating, recreation, and dog walking in these parks. In addition, multiple lines of evidence demonstrate that the cyanobacterial populations in both lakes were strongly N limited, and that the infusion of drinking water enriched in N and P may promote these events and specifically promote N limitation. Collectively, these findings provide new insight into the role of N and drinking water supplements in promoting CHABs in lakes in urban settings.

### 3.1. Dynamics of Cyanobacteria and Toxins 

During this study, near shore monitoring revealed the chronic presence of dense and toxic cyanobacteria blooms in both The Lake in Central Park and Prospect Park Lake. Cyanobacterial concentrations were often above 1 × 10^6^ µg cyanobacterial chlorophyll *a* L^−1^ and microcystin levels at times approached 1 × 10^4^ µg L^−1^. While the spatial trends of high levels of cyanobacterial densities in nearshore environments is consistent with previous work in other eutrophic freshwater systems [18,19,20,21,22], the absolute levels quantified were among the highest reported. One study, which analyzed microcystin values in 246 lakes across Canada found a maximum value of 2,153 µg L^−1^, though a majority of lakes studied (59%) were below the World Health Organization’s drinking water quality guideline of 1.0 µg L^−1^ [20]. Other studies have found high concentrations of microcystin in ephemerally formed nearshore scums: 7300 µg g^−1^ dry weight (dw) in China [21], 7100 µg g^−1^ in Portugal [22], 4100 µg g^−1^ dw in Australia [17]. In contrast, microcystin levels reported here were consistently elevated throughout the summer sampling periods for five years and were for water samples, not scums. The US EPA suggests a guidance for levels of microcystin for recreational activities in lakes at 8 µg L^−1^, and the World Health Organization (WHO) guidelines warn of a moderate probability of adverse health effects at 20 µg L^−1^ [23,24]. The NY DEC issues high toxin bloom notifications for blooms with microcystin greater than 20 µg L^−1^. Microcystin levels were, therefore, consistently orders of magnitudes above these guideline levels, indicating that these lakes are a potential public health threat to the more than 40 million people visiting them annually.

In contrast to the 2015–2019 shoreline monitoring efforts, the 2020 sampling occurred further offshore in both lakes and revealed that while cyanobacterial blooms persisted all summer within open waters, levels were generally orders of magnitude below what was present in nearshore samples. While concentrations of microcystin were substantially lower offshore, LCP consistently had levels in the open water well above the guideline values noted above. In PPL, on the other hand, levels of microcystin in the open water were relatively low throughout the summer, peaking at just over 0.5 µg L^−1^ in mid-September. While this finding contrasts with prior near shore sampling, the differences in microcystin levels between the lakes were consistent with the differences in concentrations of the microcystin synthetase gene, *mcyE*. Microcystin synthetase gene levels can be indicative of microcystin in freshwater systems [25,26,27], and, paralleling microcystin concentrations, were found at densities in LCP ~ two orders of magnitude higher than PPL. During 2020 across both sites, there was a significant correlation between the levels of *mcyE* and concentrations of microcystin (*p* < 0.0001; Appendix A). In contrast, levels of cyanobacteria were not correlated with microcystin nor with levels of *mcyE* (Appendix A) demonstrating the utility of tracking the microcystin synthetase gene rather than levels of total cyanobacteria to track bloom toxicity [2,25,26]. *Microcystis* is the most common of the microcystin producing cyanobacteria [2] and the trends in microcystin and *mcyE* were also broadly consistent with the dynamics of cyanobacteria found in each lake. *Microcystis* was ubiquitous and dominant in LCP, present in every sample between 2015 and 2020, and was the sole identifiable cyanobacteria in 64% of samples. In contrast, PPL, contained a mixed assemblage of cyanobacteria with *Microcystis* appearing in 91% of samples but being the sole identifiable cyanobacteria in only 14%.

In contrast to the dynamics of microcystin, *Microcystis*, and *mcyE* in LCP and PPL, the *sxtA* gene involved in the synthesis of the cyanobacterial toxin, saxitoxin, was found in much higher concentrations in PPL than *mcyE*, particularly in the first half of the summer when microcystin levels were below detection levels. Far less is known about saxitoxin production by cyanobacteria in freshwater compared to microcystin, and saxitoxin is rarely detected in US lakes, like due to it rarely being measured. *Aphanizomenon,* and *Dolichospermum* are two cyanobacterial genera known to produce saxitoxin [16,28] and were present in PPL when high levels of *sxtA* were measured. Still, other studies have detected *sxtA* in freshwater and have not detected saxitoxin [29], although the reliance on ELISA over HPLC or LC/MS for the quantification of saxitoxin may have prohibited quantification of low concentrations. Further study is needed to assess the potential presence and levels of saxitoxin in both lakes studied here.

### 3.2. Nutrient Limitation of CHABs

This study used a multi-tiered approach to assess N and P limitation of cyanobacterial blooms, specifically measuring levels of nutrients, assessing nutrient ratios, and performing both nutrient enrichment and nutrient dilution experiments. In 2020, DIN:DIP ratios in both LCP and PPL were well-below the Redfield ratio (<5) throughout the sampling season and were at their lowest (<1) during July and August when the cyanobacterial blooms were at their peak. Total N: total P (TN:TP) ratios were likely reflective of cyanobacterial biomass and were close to Redfield in LCP (~16) but were closer to 10 at PPL suggesting that N limitation was stronger at PPL than LCP. Beyond nutrient ratios indicating N limitation, the absolute levels of NO_x_ and NH_4_^+^ were <2 µM for most of the summer, a level below the half-saturation constant for most bloom forming cyanobacteria [30,31] and thus potentially limiting growth rates [32,33]. Ammonium and orthophosphate are released from organically rich sediments as lake temperatures rise [34,35,36,37], but the consistently low levels of NH_4_^+^ and elevated orthophosphate in both lakes implicate rapid uptake of N, and the N limited status of cyanobacterial populations. Although *Microcystis* is not diazotrophic, it is known to outcompete other cyanobacteria in N-depleted waters due to their multiple strategies allowing them to assimilate regenerated NH_4_^+^ more efficiently than other species [2,38,39].

Concentrations of PO_4_^−3^ were higher in PPL than LCP, a finding consistent with the hypothesis that N limitation may have been stronger at PPL than LCP. Higher levels of orthophosphate in eutrophic systems tend to be associated with a mixed assemblage of cyanobacteria, and can favor *Dolichospermum*/*Anabaena* [2,40]. This is consistent with the findings from this study, as *Dolichospermum* was visually identified frequently within PPL, and the *sxtA* gene carried by saxitoxin producers such as *Dolichospermum* was detected much more frequently in PPL than in LCP.

Beyond nutrient levels and ratios, our experiments directly assessed the ability of N and P to limit and stimulate algae growth in both systems. During experiments conducted in 2020, N was found to be strongly limiting in both lake systems. N enrichment stimulated cyanobacterial biomass in 8 of 10 experiments in LCP and 9 of 10 experiments in PPL. Nutrient dilution assays, however, have been proposed as a more efficient means for assessing nutrient limitation in highly eutrophic lakes [41]. The use of this alternate experimental approach yielded highly similar outcomes to direct nutrient additions, however, as the addition of N to diluted lake water stimulated cyanobacterial biomass in 8 of 10 experiments in LCP and 5 of 10 experiments in PPL.

While P is traditionally considered to be the limiting nutrient in freshwater bodies [12,42], recent studies have found N can limit the growth of cyanobacteria in temperate lakes, especially during summer [10,14,15,43,44]. This study was clearly consistent with those findings. Beyond experimental outcomes and the ratios and absolute levels of N and P, it is notable that levels of microcystin measured in both systems in 2020 were significantly correlated with the levels of total N and the TN:TP ratio and were inversely correlated with the levels of orthophosphate (*p* < 0.05 for all; Appendix A). Similarly, levels of the *mcyE* gene were also inversely correlated with levels of orthophosphate (*p* < 0.01; Appendix A). These correlations suggest that, beyond biomass limitation, the levels and synthesis of microcystin were associated with higher levels of N but lower levels of P perhaps due to the outsized role N can play in the synthesis of this toxin [10,15].

In contrast to cyanobacteria, green algae were rarely nutrient limited in LCP and PPL, perhaps due in part to their lower biomass which may have prevented the detection of a strong signal from this group in experiments. Cyanobacteria gained dominance over green algae in both lakes during the early summer of 2020 and maintained that status throughout summer. It has been established that cyanobacteria, and *Microcystis* in particular, thrive at higher temperature waters that may inhibit green algal growth [14,45,46]. Additionally, cyanobacteria likely outcompeted green algae for nutrient acquisition in experiments [14,47,48]. Indeed, nutrient additions promoted green algae densities during only 3 out of 10 whole water and 2 out of 10 diluted experiments in LCP and failed to do so in all experiments in PPL. In addition to cyanobacteria being dominant in nutrient acquisition [14,47,48], cyanobacteria may also allelopathically inhibit green algae [49].

Both NYC lakes studied here are human-made with no natural tributaries flowing into them. Consequently, water within both lakes is replenished using drinking water supplies during summer when evaporation exceeding precipitation causes water levels to decline. To assess how this might affect cyanobacterial communities, experiments were performed using drinking water collected from each park with ‘drinking water’ treatments compared to N and P enrichments. In all five experiments performed, the addition of drinking water led to an increase in cyanobacterial populations with the increases being significantly greater than control conditions in 4 of 5 experiments. In those same experiments, N treatments that were meant to mimic the levels of N in drinking water led to a similar increase in cyanobacterial biomass during the experiments indicating it was likely that the N addition to the drinking water promoted cyanobacterial growth rates during experiments. The NYC Department of Environmental Protection [50] reported that average nitrate concentration in drinking water in 2018 was 0.13 mg L^−1^ or 10 µM, similar to the levels used in experiments (20 µM in enrichment and dilution experiments; 5–15 µM in drinking water experiments). In addition to nitrate, drinking water supplies in NYC are supplemented with high levels of orthophosphate (70 µM) to minimize the leaching of lead from pipes [51,52]. While the precise amounts of drinking water added to these New York City lakes during summer is unknown, it is possible that the supplementation of lakes with drinking water with a N:P ratio of < 0.5 during summer may promote the initiation of cyanobacterial blooms as well as N limitation of cyanobacterial populations during summer. Consistent with this hypothesis, DIN:DIP ratios declined during summer and orthophosphate levels rose in LCP during summer when evaporation and water supplementation were presumably maximal. Given that supplementing human-made lakes with public water and the practice of adding orthophosphate to drinking water supplies [51,52] are common practices, it is likely these practices influence cyanobacterial blooms in other cities across the globe world where lead pipes still carry water, even after they were banned in the US 30 years ago. Finally, quantifying the relative importance of sediment, water supplementation, and watershed-based nutrient loads would be useful to elucidate the major source of N and P inputs to both systems, and, in turn, would facilitate the identification of remedial measures to mitigate these events.

### 3.3. Conclusion

The Lake in Central Park and Prospect Park Lake experience consistent, intense, and potentially hazardous CHABs on an annual basis. Concentrated cyanobacterial biomass near shorelines may pose a significant human and animal health threat as these dense accumulations of biomass, microcystin, and perhaps other toxins occur in the regions of lakes that humans and animals are most likely to interface with. While N was a clear driver of cyanobacterial populations in both lakes systems, the primary sources of N to these urban lakes is unknown. Regardless, given the ability of drinking water to promote cyanobacterial biomass experimentally, pretreatment of the water prior to supplementation of these lakes during the summer seems warranted.

## 4. Methods

### 4.1. Field Sampling and Laboratory Analyses, 2015–2019

Beginning in June 2015, water samples were collected from PPL by staff of the Prospect Park Alliance and from LCP by staff from the Central Park Conservancy. Sampling efforts were coordinated by the New York State Department of Environmental Conservation (DEC) and NYC Department of Parks and Recreation. The samples were collected along the shoreline, at high-trafficked locations, with a goal of assessing near-shore exposure risks. Samples were sent in a temperature-controlled container to the Stony Brook - Southampton laboratory, arriving the following morning. Upon arrival, samples were analyzed immediately using a BBE (biological, biophysical, engineering) Moldenke Fluoroprobe to characterize the phytoplankton community assemblage by quantifying abundances of cyanobacteria, green algae, and brown algae based on differential fluorescence of photosynthetic accessory pigments, ascribing a portion of the total chlorophyll *a* biomass to each algal group [53]. To complement fluorometry, lake samples were also preserved in 5% Lugols iodine and observed under an inverted microscope on a Sedgewick-Rafter slide to identify the numerically most abundant bloom-forming taxon. An aliquot of whole water was frozen at −80 °C for analysis of microcystin levels utilizing Eurofins Abraxis Microcystin-LR enzyme-linked immunosorbent assay (MCLR ELISA) kits [54]. Use of whole water for toxin analyses provided a combined particulate and dissolved toxin pool measurement. Samples were analyzed following the manufacturer’s protocols which involved a triple freeze–thaw cycle and a lysis of the cells using a Eurofins Abraxis QuikLyse™ Cell Lysis kit for Microcystins/Nodularins ELISA Microtiter Plate according to the manufacturer’s instructions. Lysed samples were then analyzed with a colorimetric immunoassay using a Eurofins Abraxis Microcystins/Nodularins (ADDA) ELISA Kit according to the manufacturer’s instructions. This assay detects the ADDA functional group of microcystin molecules but does not differentiate among microcystin congeners [55]. This method provided an analytical precision of ± 2% and a 96 ± 2% recovery of spiked samples.

### 4.2. Nutrient Dilution Assays, 2018

An experiment to assess whether nitrogen or phosphorus amendments would have a differential impact on cyanobacterial growth was performed in 2018 using a modified nutrient dilution approach [41]. Nutrient dilution bioassays can be more efficient at detecting limiting nutrients in hypereutrophic systems than traditional nutrient amendment assays due to the presence of nutrients at concentrations significantly higher than what is required for growth [41]. Water from each lake was split into eight triplicate treatment groups. A control consisted of 50% lake water, 50% of a multiple ion solution (MIS) containing all of the major ions found in lake water except N and P [41]. Two groups consisted of 50% lake water, 50% MIS, and nitrate additions of 5 µM and 15 µM; two groups made of 50% lake water, 50% MIS, and orthophosphate additions of 20 µM and 68 µM; two groups made of 50% lake water, 50% MIS, and nitrate and orthophosphate additions of 5 µM N and 20 µM P and 15 µM N and 68 µM P; and a final group made of 50% lake water, and 50% drinking water collected from faucets around each respective lake. The concentrations of nitrate and phosphate were within the range measured in drinking water supplied to New York City [50]. The flasks were placed in an outdoor sea table with flow through ambient water from Old Fort Pond at the Stony Brook Southampton Marine Sciences Center and were covered with a layer of screening which reduced incoming irradiance by 33% and thus providing a light level that approximated ~0.3 m in the water column. The continual flow of water from Old Fort Pond through sea tables allowed bottles to be incubated at a temperature that was within 2 ℃ of the lakes from which water was collected. Vessels were incubated for 24 h after which, a BBE Moldenke Fluoroprobe was used to characterize the phytoplankton community assemblage in each treatment group.

### 4.3. Field Sampling, 2020

Every other week from June through October 2020, 20 L water samples were obtained at sites, located offshore near central locations in each lake. During sampling, water temperature and dissolved oxygen were measured using a handheld YSI ProPlus sonde. The samples taken from each lake were then put on ice and returned to the Stony Brook University Southampton lab. Within three hours of collection, Six sets of triplicate sub-samples were obtained for multiple analyses. The phytoplankton community assemblage was characterized using a BBE Moldenke Fluoroprobe. Samples for DNA-based analysis of microcystin-producing and saxitoxin-producing cyanobacteria were obtained by filtering 10–100 mL onto a 0.22 µm polycarbonate filter. Filters were then immediately stored at −80 °C until DNA was extracted and samples were processed as described below. Whole water samples (7 mL) were frozen for later ELISA analysis of microcystins utilizing Eurofins Abraxis MCLR ELISA kits as described above. Samples were also preserved in 5% Lugols iodine and observed under an inverted microscope on a Sedgewick-Rafter slide to identify the dominate bloom-forming taxa. Nutrient levels were determined from 20 mL samples filtered through a 0.2 µm polycarbonate filter and analyzed for nitrate, ammonium, and orthophosphate, while unfiltered samples were analyzed for total nitrogen (TN), and total phosphorus (TP) on a Lachat Instruments autosampler using standard wet chemistry [56].

### 4.4. Nutrient Dilution/Amendment Assay

Experiments were performed to establish a nutrient enrichment and nutrient dilution bioassays. Using triplicate, 250 mL polycarbonate bottles, nutrient enrichment experiments were established using whole lake water that was left unamended as a control, or amended with 20 µM nitrate, 2 µM orthophosphate or both nutrients to assess which nutrient limits growth. Nutrient dilution bioassays were also established using a modified approach from Paerl and Bowles [41]. Using triplicate, 250 mL polycarbonate bottles for each treatment, lake water was diluted by 50% with MIS. The 50% MIS was not further amended as a control, whereas for three treatments, 50% lake water solution were amended with 20 µM nitrate, 2 µM orthophosphate or both nutrients to assess which nutrient limits growth. These nutrient additions are representative of pulses of nitrate and phosphate that have been previously described in eastern US freshwater systems [57,58]. Experimental vessels from both types of experiments were incubated as describe above for 24 h at light and temperature levels similar to those found at the lakes at the time of collection. After 24 h, each bottle was analyzed on a BBE Moldenke Fluoroprobe. 

### 4.5. Molecular Analysis

For molecular analysis, total nucleic acids were extracted from field samples following the methodology developed by Phytoxigene™ for the detection and quantitation of cyanobacteria and their toxin producing genes [59]. Briefly, between 5 and 100 mL of sample was concentrated on 0.2 μL polycarbonate filters then placed in BioGx Bead Lysis tubes, containing 500 µL of lysis buffer along with 300 mg of 0.1 µm glass beads and vortexed for 15 min. The solution was briefly centrifuged, then the lysate was extracted from the tubes and stored at −80 ℃ until qPCR analysis. 

The toxin-producing cyanobacterial community assemblage was assessed via qPCR and the methodology developed by Phytoxigene™ for their CyanoDTec assay, using the primer/probe sets generated by Al-Tebrineh et al. [59]. Standard curves were generated by analyzing five standards containing 20, 200, 2000, 20,000, and 200,000 copies of each target gene per µL. The 16S rRNA gene was amplified using the cyanobacteria-specific primer set developed to quantify all cyanobacterial genera. The *mcyE* and *sxtA* genes were amplified to quantify the abundance of the microcystin and saxitoxin-producing cyanobacteria. Cycling conditions were as follows: 95 ℃ for 2 min, followed by 40 cycles of 95 ℃ for 15 s and 60 ℃ for 45 s.

Two measures were taken to detect any inhibition that may occur during analysis. The 16S rRNA master mix contains an internal amplification control able to detect any inhibitors that would invalidate the assay. In addition, a sample was taken from cultures of *Microcystis* LE3 and *Dolichospermum circinale* ACBU02 and cellular density quantified. The sample was then diluted to four levels of cell concentrations to approximate levels found within the standards, following protocol developed by prior studies [60]. DNA was then extracted from these samples and analyzed both individually and in a mixed well to ensure no loss of volume corrected detection across dilutions. 

### 4.6. Statistical Analysis

A two-way analysis of variance (ANOVA) was used to assess the effects of N and P as treatments in each type of experiment and post hoc Tukey tests were used to assess differences between groups. A Spearman Rank Order Correlation matrix was used to examine relationships between environmental parameters measured in 2020 in both systems. All statistical analyses were run through R [61].

## Figures and Tables

**Figure 1 toxins-14-00684-f001:**
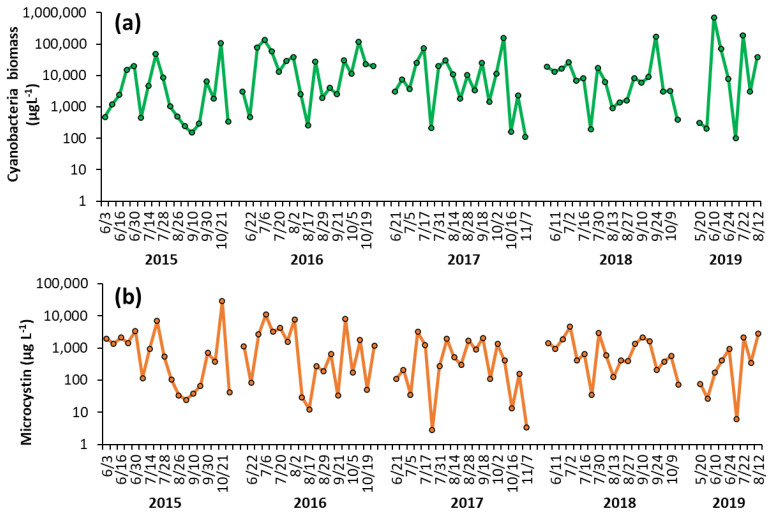
Time series of (**a**) cyanobacterial biomass and (**b**) microcystin concentration in The Lake in Central Park during the summer months from 2015–2019. Replicated measurements of cyanobacteria provided SD too small to be seen on the figure. Microcystin measurements were not replicated.

**Figure 2 toxins-14-00684-f002:**
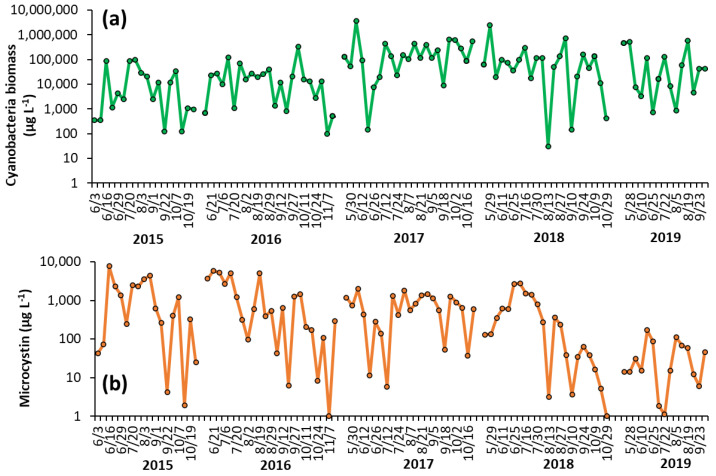
Time series for (**a**) cyanobacterial biomass and (**b**) microcystin concentration in Prospect Park Lake during the summer months from 2015–2019. Replicated measurements of cyanobacteria provided SD too small to be seen on the figure. Microcystin measurements were not replicated.

**Figure 3 toxins-14-00684-f003:**
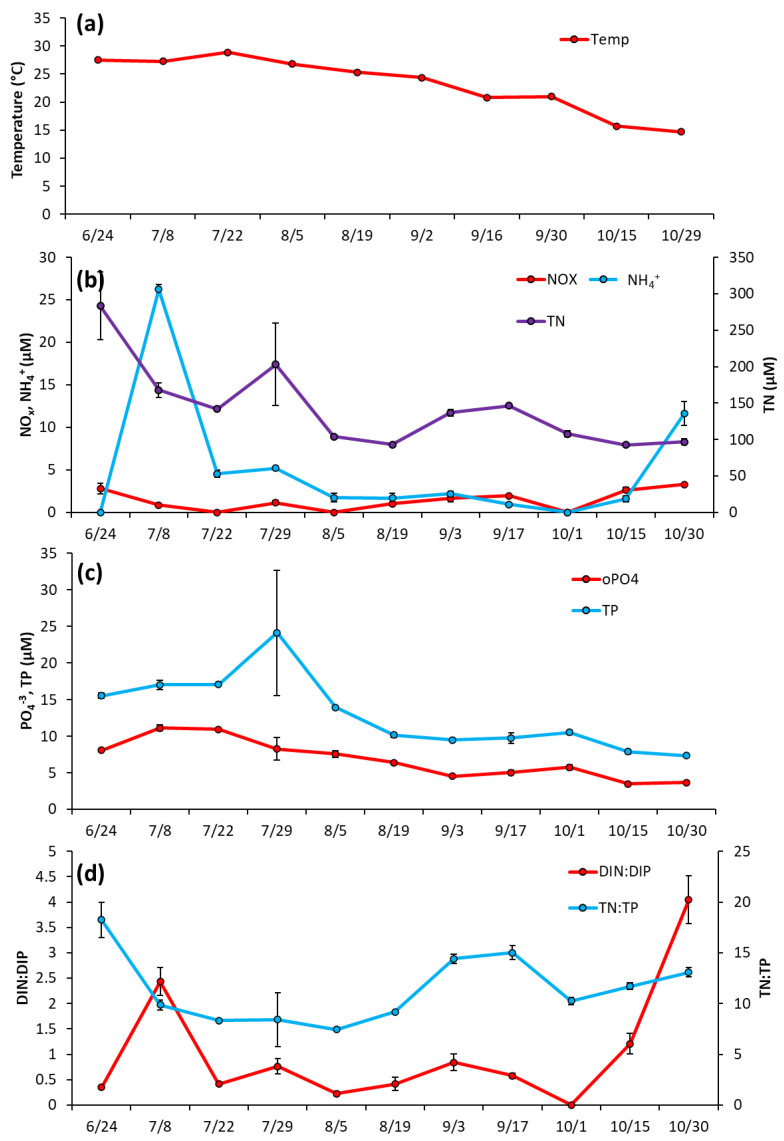
The Lake in Central Park between June and October 2020, (**a**) Surface water temperature (°C), (**b**), levels of NO_x_, NH_4_^+^ and TN (**c**), levels of TP and PO_4_^−3^ and (**d**), ratios of dissolved inorganic nitrogen to dissolved inorganic phosphorus and total nitrogen to total phosphorus. Error bars are standard deviations (*n* = 3). Non-visible error bars were smaller than data points.

**Figure 4 toxins-14-00684-f004:**
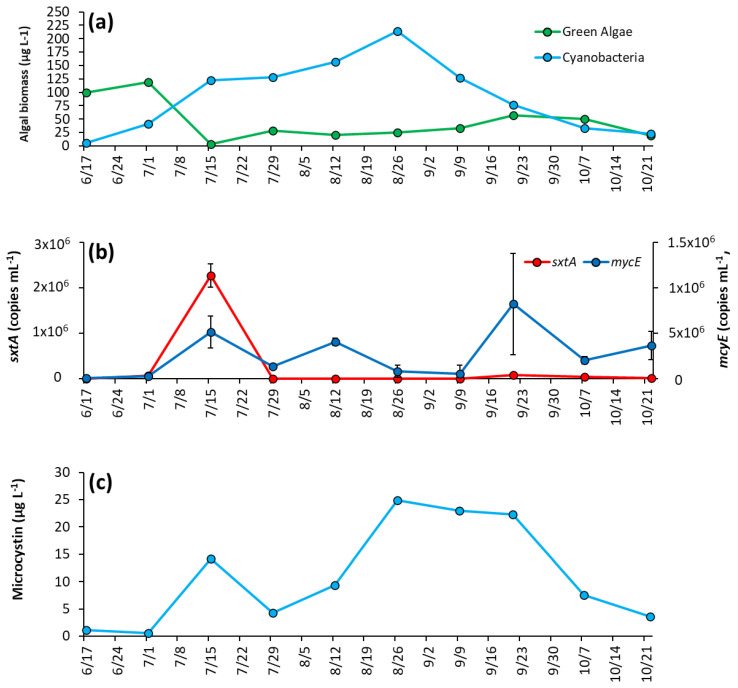
The Lake in Central Park between June and October 2020 (**a**) algal biomass, (**b**) concentration of *sxtA* and *mcyE* genes, and (**c**) microcystin concentration. Error bars are standard deviations (*n* = 3). Non-visible error bars were smaller than data points. Microcystin measurements were not replicated.

**Figure 5 toxins-14-00684-f005:**
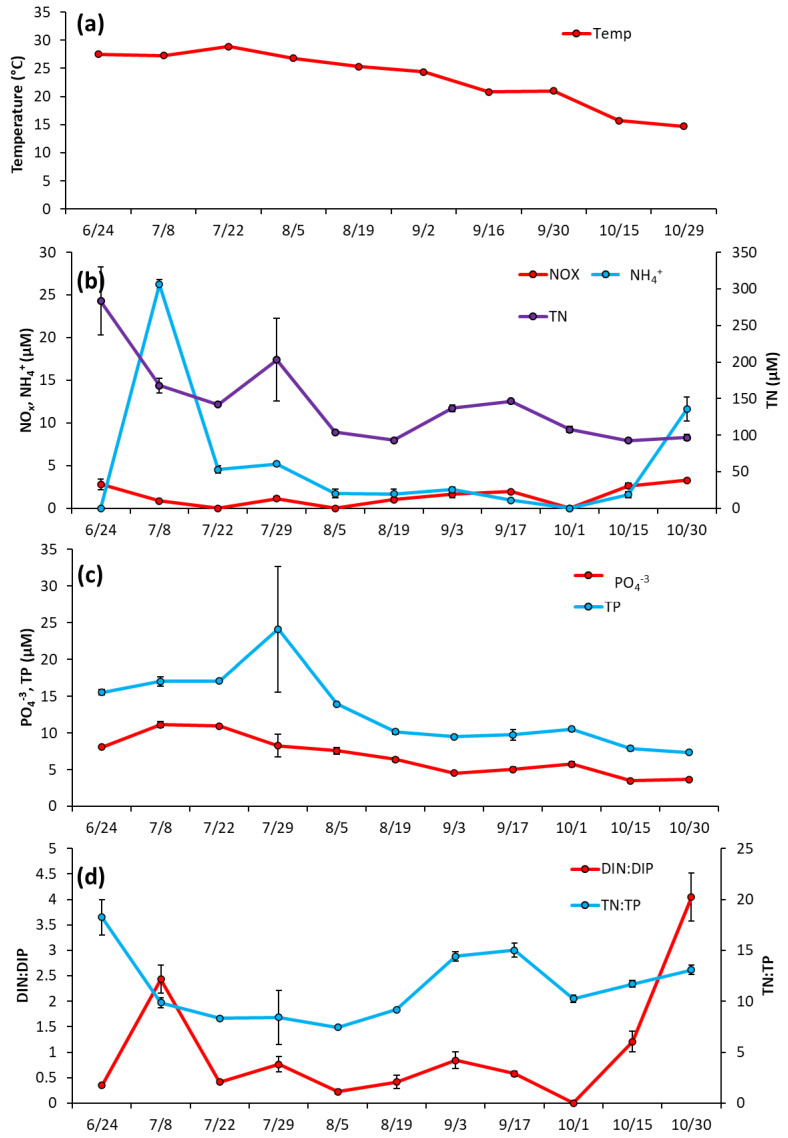
Prospect Park Lake between June and October 2020, (**a**) levels of surface temperature (°C) in the lake, (**b**), levels of NO_x_, NH_4_^+^ and TN (**c**), levels of TP and PO_4_^−3^ and (**d**), ratios of dissolved inorganic nitrogen to dissolved inorganic phosphorus and total nitrogen to total phosphorus. Error bars are standard deviations (*n* = 3). Non-visible error bars were smaller than data points.

**Figure 6 toxins-14-00684-f006:**
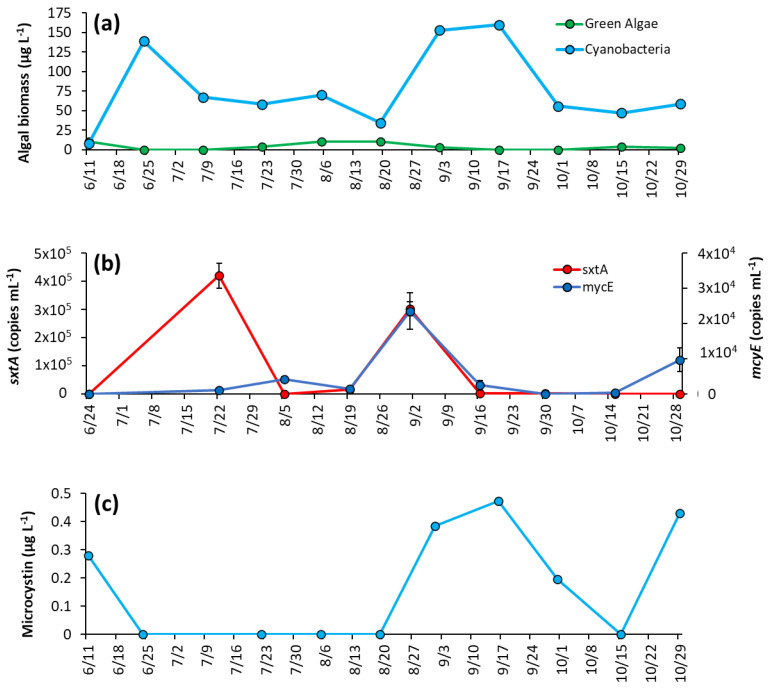
Prospect Park Lake between June and October 2020: (**a**) algal biomass, (**b**) concentration of *sxtA* and *mcyE* genes, and (**c**) microcystin concentrations. Error bars are standard deviations (*n* = 3). Non-visible error bars were smaller than data points. Microcystin measurements were not replicated.

**Figure 7 toxins-14-00684-f007:**
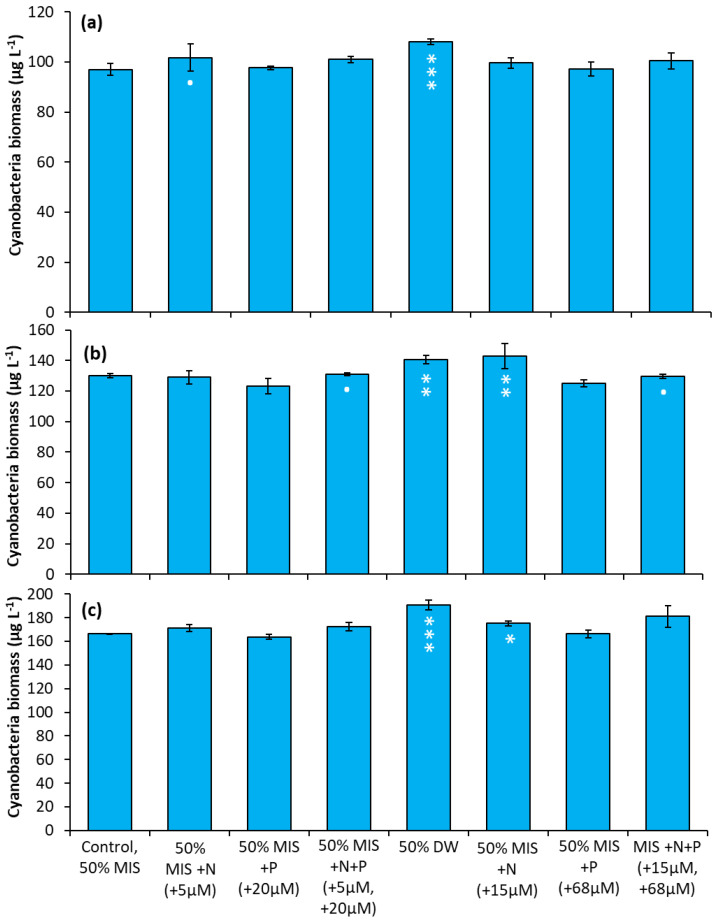
Nutrient amendment experiment utilizing water from The Lake in Central Park. (**a**) 8/22/2018. (**b**) 10/12/2018. (**c**) 10/3/2019. MIS = Multiple ion solution, N = nitrate, P = phosphorus, DW = drinking water. Precise concentrations of N and P added listed in parentheses below treatments; levels used were within the range found in NYC drinking water supplies [17]. Significance levels relative to control treatments reported based on the results of Tukey HSD tests: ‘***’ = *p*< 0.001 ‘**’ = *p* < 0.01 ‘*’ = *p* < 0.05. Error bars are standard deviations (*n* = 3).

**Figure 8 toxins-14-00684-f008:**
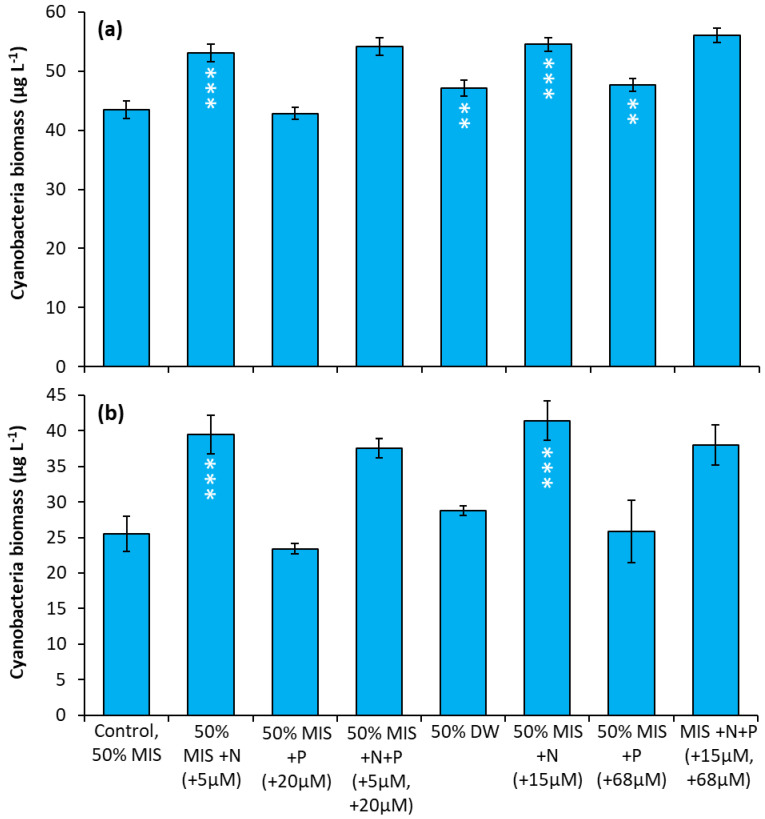
Nutrient amendment experiment utilizing water from Prospect Park Lake. **(a)** 8/22/2018. **(b)** 9/27/2018. MIS = Multiple ion solution, N = nitrate, P = phosphorus, DW = drinking water. Precise concentrations of N and P added listed in parentheses below treatments; levels used were within the range found in NYC drinking water supplies [17]. Significance levels relative to control treatments reported based on the results of Tukey HSD tests: ‘***’= *p* < 0.001, ‘**’= *p* < 0.01. Error bars are standard deviations (*n* = 3).

**Figure 9 toxins-14-00684-f009:**
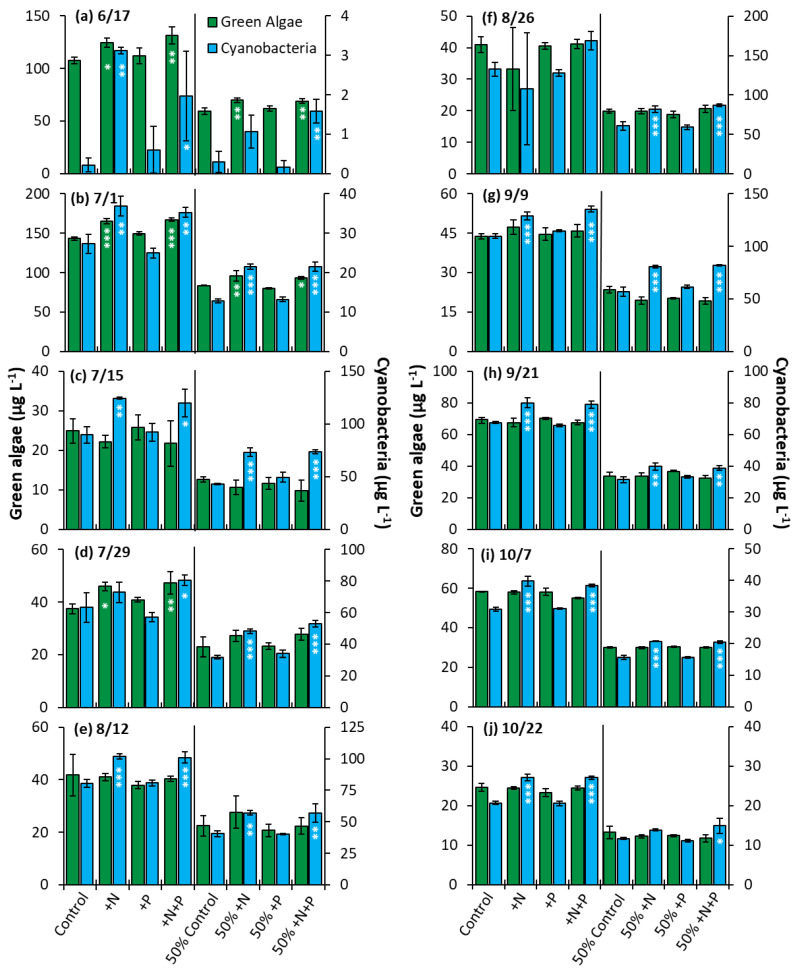
Green algae and cyanobacterial biomass in The Lake in Central Park experiments split into two groups, undiluted and 50% diluted with a Major Ion Solution (MIS), (N = nitrogen; P = phosphorus). Significance reported based on the results of Tukey’s HSD: ‘***’ = *p* < 0.001. ‘**’ = *p* < 0.01, ‘*’ = *p* < 0.05. Subfigures (**a**–**j**) are chronological order of the 2020 experiments. Error bars are standard deviations (*n* = 3).

**Figure 10 toxins-14-00684-f010:**
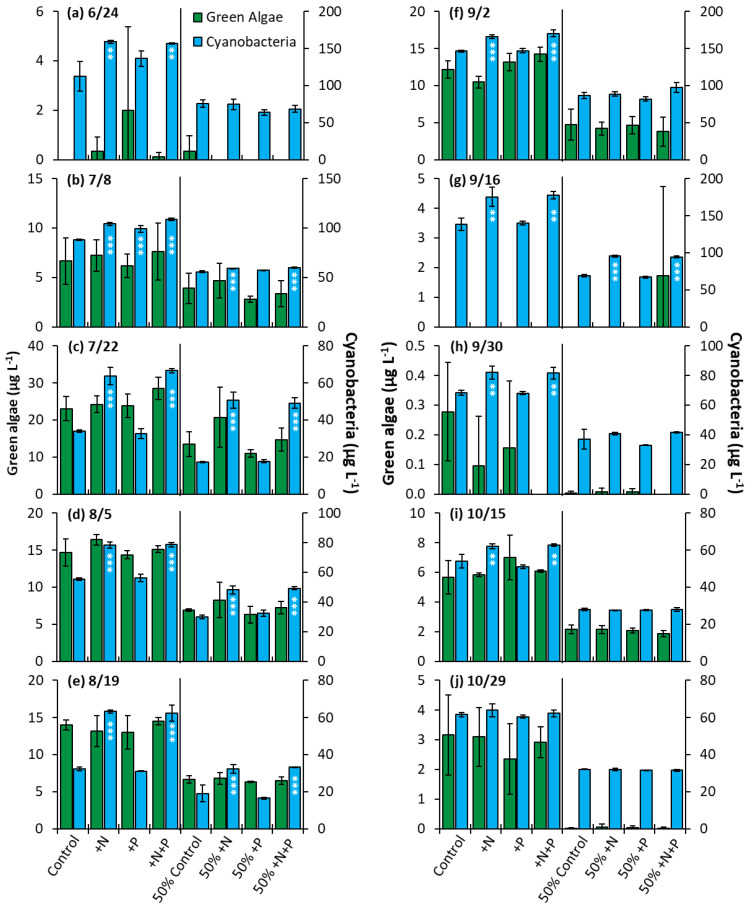
Green algae and cyanobacterial biomass in Prospect Park Lake experiments split into two groups, undiluted and 50% diluted with a Major Ion Solution (MIS), (N, nitrogen; P, phosphorus). Significance reported based on the results of Tukey’s HSD ‘***’ = *p*< 0.001. ‘**’ = *p* < 0.01. Subfigures (**a**–**j**) are chronological order of the 2020 experiments. Error bars are standard deviations (*n* = 3).

**Table 1 toxins-14-00684-t001:** Mean chlorophyll a (µg L^−1^) from cyanobacteria, green algae, and brown algae (diatoms and dinoflagellates), measured between 2015–2019 in LCP, Central Park, and PPL, Prospect Park.

	Cyanobacteria	LCP Green Algae	Brown Algae	Cyanobacteria	PPL Green Algae	Brown Algae
2015	27,500 ± 16,400	0.00	0.00	14,500 ± 5500	4.5 ± 2.2	0.65 ± 0.40
2016	37,600 ± 10,500	0.00	0.00	28,200 ± 12,200	2.9 ± 2.3	0.00
2017	21,500 ± 9060	0.00	0.00	327,000 ± 136,000	0.43 ± 0.40	0.00
2018	17,600 ± 10,500	0.59 ± 0.59	0.00	210,000 ± 110,000	0.02± 0.02	0.00
2019	11,200 ± 7650	16.5 ± 11.8	0.33 ± 2.00	134,000 ± 56,600	2.7 ± 2.7	3.9 ± 3.9

## Data Availability

Data is contained within the article and Appendix A.

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
