# Peer review of "Nitrogen Limitation of Intense and Toxic Cyanobacteria Blooms in Lakes within Two of the Most Visited Parks in the USA: The Lake in Central Park and Prospect Park Lake"

_toxins, 2022, doi:10.3390/toxins14100684_

Round 1
Reviewer 1 Report
This is a neat study examining blooms in 2 lakes in NY with cyanobacterial blooms. I have only minor suggestions and once these are addressed the paper can be accepted.
As cyanobacterial pigment concentration is used instead of counts/biovolume, it would be good to give the reader an indication of what different concentrations correspond to approximately to counts.
In the discussion and talk about green algae only responding to 3 out of 10 or 2 out of 10 experiments, it should be remembered that no response compared to the control may mean they are not limited by nutrients - which could be an advantage over algae that are limited.
Results - no need to give the numbers on text of the seasonal biomass - this is shown in graphs.
Tables 2, 3, 4 should be in supplementary data - not in main manuscript.
conclusion should be just the main findings. Put ideas for future research into the discussion.
Author Response
Response to Review #1:
Comments in italics, response in bold
The manuscript entitled ‘Nitrogen limitation of intense and toxic cyanobacteria blooms in two of the most visited parks in the US: The Lake in Central Park and Prospect Park Lake’ describes the results of the field and experimental studies on cyanobacteria causing water blooms in two lakes in the USA. I find this work well-written and interesting, however, the authors should use the Microsoft Word template from https://www.mdpi.com/journal/toxins/instructions. The English language is good. I have some suggestions for the introduction, methods, and presentation of the results as well as other minor, mostly editorial, comments.
We thank the reviewer for their comments.
I would also recommend changing the title to ‘Nitrogen limitation of intense and toxic cyanobacteria blooms in water bodies located in two of the most visited parks in the US: The Lake in Central Park and Prospect Park Lake’
This change was made.
Introduction
Add information about the potential negative impact of cyanobacteria blooms on human health (skin allergies, inhalation of aerosol containing cyanobacterial cells, etc.).
We have added the text: “…with human health impairment associated with skin allergies, inhalation of aerosol containing cyanobacterial cells, and consumption of drinking water tainted with cyanotoxins.”
Please, standardize the spelling of CHABs throughout the text.
This change has been made
Remove unnecessary spaces.
We have searched for and removed unnecessary spaces, partly by left margin justifying all text.
Methods
subtitles - replace with “Field sampling and laboratory analyzes”
This change has been made.
Were cyanobacteria counted to find out which taxa were dominants? The Authors write about cyanobacterial and green algal biomass, but there is no information about the method of biomass measurement. I understand that chlorophyll is the equivalent of biomass but it has to be clearly written in the methods.
To address this issue, we have updated the methods to provide clarity on this issue. The methods text now reads: “Upon arrival, the samples were analyzed immediately using a BBE Moldenke Fluoroprobe that quantifies abundances of cyanobacteria, green algae, and brown algae based on differential fluorescence of photosynthetic accessory pigments and ascribes a portion of the total chlorophyll a to each algal group [15,17] was used to characterize the phytoplankton community assemblage in each treatment group. To complement fluorometry, lake samples were also preserved in 5% Lugols iodine and observed under an inverted microscope on a Sedgewick-Rafter slide to identify the dominate bloom-forming taxa.”
Please, clearly characterize the temperature value, the light intensity, and the day/night exposure time during the experiment.
To address this issue we have updated the methods text to provide clarity on this issue. The methods now read: “The flasks were placed in an outdoor sea table with flow through ambient water from Old Fort Pond at the Stony Brook Southampton Marine Sciences Center and were covered with a layer of screening which reduced incoming irradiance by 33% and thus providing a light level that approximated ~0.3 m in the water column. The continual flow of water from Old Fort Pond through sea tables allowed bottles to be incubated at a temperature that was within 2℃ of the lakes from which water was collected.”
Describe how samples were prepared for toxin analyses (were they extracted? How? What kind of toxins (extracellular, intracellular, or both) were analyzed?
The following text was added to the methods:
“An aliquot of whole water was frozen at -80°C for later analysis of microcystin levels utilizing Eurofins Abraxis Microcystin-LR enzyme-linked immunosorbent assay (MCLR ELISA) kits. Use of whole water for toxin analyses provided a combined particulate and dissolved toxin pool measurement. Samples were analyzed following the manufacturer’s protocols which involved a triple freeze-thaw cycle and a lysis of the cells using an Eurofins Abraxis QuikLyse™ Cell Lysis kit for Microcystins/Nodularins ELISA Microtiter Plate according to the manufacturer’s instructions. Lysed samples were then analyzed with a colorimetric immunoassay using an Eurofins Abraxis Microcystins/Nodularins (ADDA) ELISA Kit according to the manufacturer’s instructions. This method provided an analytical precision of ±2 % and a 96 ± 2 % recovery of spiked samples.”.
Results
In tables 2, 3 and 4, there is no Anabaena, whereas this genus is mentioned in the text a few times. Of course, planktonic Anabaena is now named Dolichospermum.
All references to Anabaena in the text were changed to Dolichospermum.
I would recommend changing the information in tables – remove everywhere “sp.” as it suggests that there was always only one taxon from particular genera, or live “sp.” (but not in italics), and change the title of columns from “Cyanobacteria genera” to “cyanobacterial taxa”.
These changes have been made.
Page 7, second line from the top of the text – change “was” to “were”
Page 7 was in the methods; we searched for but could not find the ‘was’ that is being referred to here, but would still like to change it.
Figures 1-3 and their captions – add letter markings in graphs, captions, and in the text (like in the case of other figures)
These changes have been made.
Standardize the titles of axes – for example, Fig. 4 a – there is chlorophyll a, whereas in other figures there is biomass.
These have ben standardized to ‘biomass’.
Page 10, first line – add “concentration”: “The saxitoxin biosynthesis gene, sxtA, concentration…”. Change the font in the next paragraph.
This change has been made.
Discussion
“Dynamics of cyanobacteria and toxins” the second row – “The Lake”
This change was made.
References – write the Latin names in italics
All genus and species Latin names were italicized.
Reviewer 2 Report
The manuscript entitled ‘Nitrogen limitation of intense and toxic cyanobacteria blooms in two of the most visited parks in the US: The Lake in Central Park and Prospect Park Lake’ describes the results of the field and experimental studies on cyanobacteria causing water blooms in two lakes in the USA. I find this work well-written and interesting, however, the authors should use the Microsoft Word template from https://www.mdpi.com/journal/toxins/instructions. The English language is good. I have some suggestions for the introduction, methods, and presentation of the results as well as other minor, mostly editorial, comments.
I would also recommend changing the title to ‘Nitrogen limitation of intense and toxic cyanobacteria blooms in water bodies located in two of the most visited parks in the US: The Lake in Central Park and Prospect Park Lake’
Introduction
Add information about the potential negative impact of cyanobacteria blooms on human health (skin allergies, inhalation of aerosol containing cyanobacterial cells, etc.).
Please, standardize the spelling of CHABs throughout the text. Remove unnecessary spaces.
Methods
subtitles - replace with “Field sampling and laboratory analyzes”
Were cyanobacteria counted to find out which taxa were dominants? The Authors write about cyanobacterial and green algal biomass, but there is no information about the method of biomass measurement. I understand that chlorophyll is the equivalent of biomass but it has to be clearly written in the methods.
Please, clearly characterize the temperature value, the light intensity, and the day/night exposure time during the experiment.
Describe how samples were prepared for toxin analyses (were they extracted? How? What kind of toxins (extracellular, intracellular, or both) were analyzed?
Results
In tables 2, 3 and 4, there is no Anabaena, whereas this genus is mentioned in the text a few times. Of course, planktonic Anabaena is now named Dolichospermum.
I would recommend changing the information in tables – remove everywhere “sp.” as it suggests that there was always only one taxon from particular genera, or live “sp.” (but not in italics), and change the title of columns from “Cyanobacteria genera” to “cyanobacterial taxa”.
Page 7, second line from the top of the text – change “was” to “were”
Figures 1-3 and their captions – add letter markings in graphs, captions, and in the text (like in the case of other figures)
Standardize the titles of axes – for example, Fig. 4 a – there is chlorophyll a, whereas in other figures there is biomass.
Page 10, first line – add “concentration”: “The saxitoxin biosynthesis gene, sxtA, concentration…”. Change the font in the next paragraph.
Discussion
“Dynamics of cyanobacteria and toxins” the second row – “The Lake”
References – write the Latin names in italics
Author Response
Response to Review #2:
Comments in italics, response in bold
This study performed 5 years of nearshore sampling of the Lake in Central Park (LCP) and Prospect Park Lake (PPL) in New York City (NYC), USA, two of the most visited parks in the US. Also, nutrient addition and dilution experiments were conducted to assess whether N, P, a combination of both elements, or neither were limiting to the various phytoplankton populations. Concentrations of microcystin and concentrations of microcystin and saxitoxin synthesis genes were also detected. The data is interesting. However, I have the following comments and suggestions for the authors to improve the quality of manuscript.
We thank the reviewer for their comments.
- Figures 1 and 2
How did you calculate the cyanobacterial biomass? Please add the text in the section of Methods.
To address this issue, we have updated the methods to provide clarity on this issue. The methods text now reads: “Upon arrival, the samples were analyzed immediately using a BBE Moldenke Fluoroprobe that quantifies abundances of cyanobacteria, green algae, and brown algae based on differential fluorescence of photosynthetic accessory pigments and ascribes a portion of the total chlorophyll a to each algal group [15,17] was used to characterize the phytoplankton community assemblage in each treatment group. To complement fluorometry, lake samples were also preserved in 5% Lugols iodine and observed under an inverted microscope on a Sedgewick-Rafter slide to identify the dominate bloom-forming taxa.”
- “Throughout the study of LCP and PPL, cyanobacteria dominated phytoplankton biomass making up 89.5% of total chlorophyl in 2015, 83.6% in 2016, 96.6% in 2017, 94.5% in 2018, and 99.4% in 2019 in LCP (Table 1), and 91.5% in 2015, 86.2% in 2016, 97.1% in 2017, 99.2% in 2018, and 99.9% in 2019 in PPL (Table 1).”
Table 1
It seems that there are chlorophyll a (μ g L-1) from organisms other than cyanobacteria, green algae, and brown algae (diatoms and dinoflagellates). Please insert the data in the Table 1.
The values here are correct but the percentages are wrong. We have corrected the percentages in the text.
- Tables 2 and 3
How do you define most abundant cyanobacteria? Please add the definition in the revised manuscript.
The methods now state: “Samples were also preserved in 5% Lugols iodine and observed under an inverted microscope on a Sedgewick-Rafter slide to identify the dominate bloom-forming taxa with the most numerically abundant taxon noted.”
- “In LCP toxin concentrations averaged 2.73 ± 1.58 x 103 μg L-1 in 2015, 2.32 ± 0.75 x 103 μg L-1 in 2016, 7.34 ± 2.09 x 102 μg L-1 in 2017, 1.09 ± 0.27 x 103 μg L-1 in 2018, and 7.70 ± 3.42 x 102 μg L-1 in 2019 (Figure 1).”
Figures 1, 2, 4c and 6c
The concentrations of MCs are extracellular, intracellular or total? Please add the descriptions in the section of Methods. Also, More than 279 derivatives of MCs have been reported. Method by enzyme-linked immunosorbent assay (ELISA) cannot determine the derivatives of MCs. See and cite the following paper. Please add some discussion for the limitations.
Challenges of using blooms of Microcystis spp. in animal feeds: A comprehensive review of nutritional, toxicological and microbial health evaluation. https://doi.org/10.1016/j.scitotenv.2020.142319
We have clarified this point by altering the description of this method as follows:
“An aliquot of whole water was frozen at -80°C for later analysis of microcystin levels utilizing Eurofins Abraxis Microcystin-LR enzyme-linked immunosorbent assay (MCLR ELISA) kits. Use of whole water for toxin analyses provided a combined particulate and dissolved toxin pool measurement. Samples were analyzed following the manufacturer’s protocols which involved a triple freeze-thaw cycle and a lysis of the cells using a Eurofins Abraxis QuikLyse™ Cell Lysis kit for Microcystins/Nodularins ELISA Microtiter Plate according to the manufacturer’s instructions. Lysed samples were then analyzed with a colorimetric immunoassay using a Eurofins Abraxis Microcystins/Nodularins (ADDA) ELISA Kit according to the manufacturer’s instructions. This assay detects the ADDA functional group of microcystin molecules but does not differentiate among microcystin congeners [18]. This method provided an analytical precision of ±2 % and a 96 ± 2 % recovery of spiked samples.”
Reference 18 is now Challenges of using blooms of Microcystis spp. in animal feeds: A comprehensive review of nutritional, toxicological and microbial health evaluation. https://doi.org/10.1016/j.scitotenv.2020.142319
- “The Lake in Central Park – Over most of the summer (June 16 – Sept 9) temperature levels remained fairly consistent (25.8 ± 1.7 °C), before dropping to ~17 °C by 9/21 and remaining around that level for the rest of the season (Figure 3a).”
Figure 3
“June 16”? In the Figure 3, the date is June 17. Please check it.
Changed to June 17.
Please change “9/21” to “Sept/21”.
Changed as requested.
- Figures 4 and 6
Concentration of sxtA and mcyE genes were detected, but only concentrations of microcystins were measured.
Yes, that is correct.
- Section “Nutrient amendment experiments using drinking water and nutrients, 2018”
Please put this section before section “Nutrient amendment experiments using drinking water and nutrients, 2018”.
This change has been made.
- Please add correlation analysis and multivariate statistical analysis for relationship of cyanobacteria/cyanotoxins and environmental factors, including nutrients N and P.
We performed a Spearmans correlation analysis and did find some interesting correlations regarding microcystin, mcyE, and cyanobacteria, as well as among microcystin, mcyE and N and P pools. We have added this information and interpreted it within the Discussion. The correlation matrix is now Supplemental Table 4
- Why cyanotoxins were not analyzed in nutrient addition and dilution experiments?
We did not collect these samples. In hindsight, we would have liked to have had these samples.
- Discussion
“In contrast to the dynamics of microcystin, Microcystis, and mcyE in LCP and PPL, the sxtA gene involved in the synthesis of the cyanobacterial toxin, saxitoxin, was found in much higher concentrations in PP than mcyE, particularly in the first half of the summer when microcystin levels were below detection levels.”
What is “PP”?
This should have been PPL and has been changed to PPL.
Reviewer 3 Report
Journal: Toxins (ISSN 2072-6651)
Manuscript ID: toxins-1925304
Type: Article
Title: Nitrogen limitation of intense and toxic cyanobacteria blooms in two of the most visited parks in the US: The Lake in Central Park and Prospect Park Lake
Section: Marine and Freshwater Toxins
This study performed 5 years of nearshore sampling of the Lake in Central Park (LCP) and Prospect Park Lake (PPL) in New York City (NYC), USA, two of the most visited parks in the US. Also, nutrient addition and dilution experiments were conducted to assess whether N, P, a combination of both elements, or neither were limiting to the various phytoplankton populations. Concentrations of microcystin and concentrations of microcystin and saxitoxin synthesis genes were also detected. The data is interesting. However, I have the following comments and suggestions for the authors to improve the quality of manuscript.
1. Figures 1 and 2
How did you calculate the cyanobacterial biomass? Please add the text in the section of Methods.
2. “Throughout the study of LCP and PPL, cyanobacteria dominated phytoplankton biomass making up 89.5% of total chlorophyl in 2015, 83.6% in 2016, 96.6% in 2017, 94.5% in 2018, and 99.4% in 2019 in LCP (Table 1), and 91.5% in 2015, 86.2% in 2016, 97.1% in 2017, 99.2% in 2018, and 99.9% in 2019 in PPL (Table 1).”
Table 1
It seems that there are chlorophyll a (μ g L-1) from organisms other than cyanobacteria, green algae, and brown algae (diatoms and dinoflagellates). Please insert the data in the Table 1.
3. Tables 2 and 3
How do you define most abundant cyanobacteria? Please add the definition in the revised manuscript.
4. “In LCP toxin concentrations averaged 2.73 ± 1.58 x 103 μg L-1 in 2015, 2.32 ± 0.75 x 103 μg L-1 in 2016, 7.34 ± 2.09 x 102 μg L-1 in 2017, 1.09 ± 0.27 x 103 μg L-1 in 2018, and 7.70 ± 3.42 x 102 μg L-1 in 2019 (Figure 1).”
Figures 1, 2, 4c and 6c
The concentrations of MCs are extracellular, intracellular or total? Please add the descriptions in the section of Methods. Also, More than 279 derivatives of MCs have been reported. Method by enzyme-linked immunosorbent assay (ELISA) cannot determine the derivatives of MCs. See and cite the following paper. Please add some discussion for the limitations.
Challenges of using blooms of Microcystis spp. in animal feeds: A comprehensive review of nutritional, toxicological and microbial health evaluation. https://doi.org/10.1016/j.scitotenv.2020.142319
5. “The Lake in Central Park – Over most of the summer (June 16 – Sept 9) temperature levels remained fairly consistent (25.8 ± 1.7 °C), before dropping to ~17 °C by 9/21 and remaining around that level for the rest of the season (Figure 3a).”
Figure 3
“June 16”? In the Figure 3, the date is June 17. Please check it.
Please change “9/21” to “Sept/21”.
6. Figures 4 and 6
Concentration of sxtA and mcyE genes were detected, but only concentrations of microcystins were measured.
concentration
7. Section “Nutrient amendment experiments using drinking water and nutrients, 2018”
Please put this section before section “Nutrient amendment experiments using drinking water and nutrients, 2018”.
8. Please add correlation analysis and multivariate statistical analysis for relationship of cyanobacteria/cyanotoxins and environmental factors, including nutrients N and P.
9. Why cyanotoxins were not analyzed in nutrient addition and dilution experiments?
10. Discussion
“In contrast to the dynamics of microcystin, Microcystis, and mcyE in LCP and PPL, the sxtA gene involved in the synthesis of the cyanobacterial toxin, saxitoxin, was found in much higher concentrations in PP than mcyE, particularly in the first half of the summer when microcystin levels were below detection levels.”
What is “PP”?
Author Response
Response to Review #3:
Comments in italics, response in bold
This is a neat study examining blooms in 2 lakes in NY with cyanobacterial blooms. I have only minor suggestions and once these are addressed the paper can be accepted.
We thank the reviewer for their feedback.
As cyanobacterial pigment concentration is used instead of counts/biovolume, it would be good to give the reader an indication of what different concentrations correspond to approximately to counts.
There are no direct comparisons as different cells contain different amounts of phycocyanin and chlorophyll a.
In the discussion and talk about green algae only responding to 3 out of 10 or 2 out of 10 experiments, it should be remembered that no response compared to the control may mean they are not limited by nutrients - which could be an advantage over algae that are limited.
This revised manuscript contained a paragraph dedicated to this topic in the Discussion:
“In contrast to cyanobacteria, green algae were rarely nutrient limited in LCP and PPL, perhaps due in part to their lower biomass which may have prevented the detection of a strong signal from this group in experiments. Cyanobacteria gained dominance over green algae in both lakes during the early summer of 2020 and maintained that status throughout summer. It has been established that cyanobacteria, and Microcystis in particular, thrive at higher temperature waters that may inhibit green algal growth [15,56,57]. Additionally, cyanobacteria likely outcompeted green algae for nutrient acquisition in experiments [15,58,59]. Indeed, nutrient additions promoted green algae densities during only 3 out of 10 whole water and 2 out of 10 diluted experiments in LCP and failed to do so in all experiments in PPL. In addition to cyanobacteria being dominant in nutrient acquisition [15,58,59], cyanobacteria may also allelopathically inhibit green algae [60].”
Results - no need to give the numbers on text of the seasonal biomass - this is shown in graphs.
Several numbers were eliminated from the results.
Tables 2, 3, 4 should be in supplementary data - not in main manuscript.
These have been made Supplemental Tables and are no longer in the main manuscript.
Conclusion should be just the main findings. Put ideas for future research into the discussion.
We agree with the reviewer and these statements were moved from the Conclusion to the Discussion.
Round 2
Reviewer 3 Report
Journal: Toxins (ISSN 2072-6651)
Manuscript ID: toxins-1925304-peer-review-v2
Type: Article
Title: Nitrogen limitation of intense and toxic cyanobacteria blooms in water bodies in located in two of the most visited parks in the US: The Lake in Central Park and Prospect Park Lake
Section: Marine and Freshwater Toxins
This study performed 5 years of nearshore sampling of the Lake in Central Park (LCP) and Prospect Park Lake (PPL) in New York City (NYC), USA, two of the most visited parks in the US. Also, nutrient addition and dilution experiments were conducted to assess whether N, P, a combination of both elements, or neither were limiting to the various phytoplankton populations. Concentrations of microcystin and concentrations of microcystin and saxitoxin synthesis genes were also detected.
The quality of manuscript has improved during revisions. However, there are still some issues need to be addressed. The detailed suggestions and comments are as follows:
1. Title
Change to “Nitrogen limitation of intense and toxic cyanobacteria blooms in water bodies located in two of the most visited parks in the US: The Lake in Central Park and Prospect Park Lake”
2. METHODS
Please draw a map figure to show the location of The Lake in Central Park and Prospect Park Lake, as well as the sampling sites.
3. Section “Field Sampling, 2020”
“Whole water samples (7 mL) were frozen for later ELISA analysis of microcystins [21] utilizing Eurofins Abraxis MCLR ELISA kits.”
Please move the citation of reference to the following sentence in the section “Field Sampling and Laboratory Analyses, 2015-2019”: “An aliquot of whole water was frozen at -80°C for analysis of microcystin levels utilizing Eurofins Abraxis Microcystin-LR enzyme-linked immunosorbent assay (MCLR ELISA) kits.”
4. Cyanobacteria or chlorophyll a concentrations
Please check the entire manuscript, tables and figures. The descriptions of cyanobacteria or chlorophyll a should be right. Although chlorophyll is the equivalent of biomass, they are not the same, especially when you describe concentrations.
5. Section “Shoreline microcystin concentrations, 2015-2019”
“Levels of microcystin in the lake varied throughout the monitoring period but was detected in nearly every sample analyzed from 2015 - 2019 (100% in LCP, 98% in PPL).”
Please change “was” to “were”.
6. Presentation of results
Standard deviations (SD) are only shown in Figures 3bc, 4bc, 5bc, 6b. How many sampling sites in each lake? Did you perform repeat experiments for other parameters? Why standard deviations (SD) are not shown in other figures? Please present the data with SD.
7. Supplemental Table 4
Please move this table to the main text. Also, describe the results in the section of “results”.
8. Supplemental Table 4
Presentation of data can be improved, e.g., please change “0.128 0.585 20” to “0.128, 0.585, 20”.
Author Response
Response to reviewer comments
Comments in italics
Respond in bold
- Title
Change to “Nitrogen limitation of intense and toxic cyanobacteria blooms in water bodies located in two of the most visited parks in the US: The Lake in Central Park and Prospect Park Lake”
This change was made.
- METHODS
Please draw a map figure to show the location of The Lake in Central Park and Prospect Park Lake, as well as the sampling sites.
This map was made and is now Supplemental Figure 1.
- Section “Field Sampling, 2020”
“Whole water samples (7 mL) were frozen for later ELISA analysis of microcystins [21] utilizing Eurofins Abraxis MCLR ELISA kits.”
Please move the citation of reference to the following sentence in the section “Field Sampling and Laboratory Analyses, 2015-2019”: “An aliquot of whole water was frozen at -80°C for analysis of microcystin levels utilizing Eurofins Abraxis Microcystin-LR enzyme-linked immunosorbent assay (MCLR ELISA) kits.”
This change was made and the references were renumbered accordingly.
- Cyanobacteria or chlorophyll a concentrations
Please check the entire manuscript, tables and figures. The descriptions of cyanobacteria or chlorophyll a should be right. Although chlorophyll is the equivalent of biomass, they are not the same, especially when you describe concentrations.
We have checked the document and made changes and paid specific attention to the method to ensure the precise parameter being measured is clear, which should make it clear through the document. If there are still specific problems with this, please let us know.
- Section “Shoreline microcystin concentrations, 2015-2019”
“Levels of microcystin in the lake varied throughout the monitoring period but was detected in nearly every sample analyzed from 2015 - 2019 (100% in LCP, 98% in PPL).”
Please change “was” to “were”.
This change was made.
- Presentation of results
Standard deviations (SD) are only shown in Figures 3bc, 4bc, 5bc, 6b. How many sampling sites in each lake? Did you perform repeat experiments for other parameters? Why standard deviations (SD) are not shown in other figures? Please present the data with SD.
In some cases, the SD presented are too small to see. In other cases, we’ve gone back and added the SD to the figures. In one case, we do not have replicated measurements: microcystin. We now indicate all of these details in all of the figure legends.
- Supplemental Table 4
Please move this table to the main text. Also, describe the results in the section of “results”.
The table is four pages and will not fit cleanly within the manuscript and contains primarily negative results (non-correlations). Since the results were divided by site and the correlation matrix was constructed by a reviewer’s request to examine trends across sites, the outcome does not fit cleanly into the results, but does nicely complement our Discussion points.
- Supplemental Table 4
Presentation of data can be improved, e.g., please change “0.128 0.585 20” to “0.128, 0.585, 20”.
This change was made.